# Loop-Mediated Isothermal Amplification for the Fast Detection of *Bonamia ostreae* and *Bonamia exitiosa* in Flat Oysters

**DOI:** 10.3390/pathogens13020132

**Published:** 2024-01-30

**Authors:** Irene Cano, Gareth Wood, David Stone, Mathilde Noyer, Lydie Canier, Isabelle Arzul

**Affiliations:** 1The International Centre of Excellence for Aquatic Animal Health, Cefas Weymouth Laboratory, Weymouth, Dorset DT4 8UB, UKdavid.stone@cefas.gov.uk (D.S.); 2The Institut Français de Recherche pour l’Exploitation de la Mer Ifremer, RBE-SG2M-ASIM, Station de La Tremblade, Avenue de Mus de Loup, La Tremblade, 17390 Brest, France; mathilde.noyer@ifremer.fr (M.N.); lydie.canier@ifremer.fr (L.C.); isabelle.arzul@ifremer.fr (I.A.)

**Keywords:** *Bonamia ostreae*, *Bonamia exitiosa*, bonamiosis, oyster, haplosporidia, LAMP, diagnostic, point-of-care test

## Abstract

The haplosporidian parasites *Bonamia ostreae* (BO) and *B. exitiosa* (BE) are serious oyster pathogens. Two independent laboratories evaluated fluorescence real-time loop-mediated isothermal amplification (LAMP) assays for rapidly detecting these parasites. Specific LAMP assays were designed on the BO *actin-1* and BE *actin* genes. A further generic assay was conceived on a conserved region of the *18S* gene to detect both *Bonamia* species. The optimal reaction temperature varied from 65 to 67 °C depending on the test and instrument. Melting temperatures were 89.8–90.2 °C, 87.0–87.6 °C, and 86.2–86.6 °C for each of the BO, BE, and generic assays. The analytical sensitivity of these assays was 50 copies/µL in a 30 min run. The BO and BE test sensitivity was ~1 log lower than a real-time PCR, while the generic test sensitivity was similar to the real-time PCR. Both the BO and BE assays were shown to be specific; however, the generic assay potentially cross-reacts with *Haplosporidium costale*. The performance of the LAMP assays evaluated on samples of known status detected positives within 7–20 min with a test accuracy of 100% for the BO and generic tests and a 95.8% accuracy for BE. The ease of use, rapidity and affordability of these tests allow for field deployment.

## 1. Introduction

The haplosporidian microcells *Bonamia ostreae* and *Bonama exitiosa* are the causative agents of bonamiosis, a notifiable disease according to the World Organization for Animal Health (WOAH) [1,2]. These protozoans parasitise different species of oysters worldwide, such as the European flat oyster (*Ostrea edulis* L.), the Chilean oyster (*Ostrea chilensis* Kűster), and the Ariake cupped oyster (*Magallana* [syn. *Crassostrea*] *ariakensis*) [1,3]. *B. exitiosa* has also been reported to infect other oyster species, such as the Australian flat oyster (*Ostrea angasi* Sowerby), the Olympia oyster (*Ostrea lurida* Carpenter), the *Ostrea puelchana* d’Orbigny, and the Suminoe oyster (*Crassostrea ariakensis* Fujita) [2,3,4,5].

*B. ostreae* was first identified in France in 1979 [6]. Quickly, the parasite was identified in other European countries such as the Netherlands, Spain, Denmark, and the UK [7,8]. It was also soon reported outside Europe in the USA [9] and, more recently, in New Zealand in 2015 (unpublished data, confirmed by the Australian Animal Health Laboratory and reported to the WHOA). Similarly, *B. exitiosa* was first described in New Zealand in the 1980s and later reported in Australia, Europe, the USA, and South America, infecting different oyster species [4,10,11,12]. The spread of *Bonamia* parasites likely occurred through multiple movements of infected oyster seeds; thus, to prevent the further spread of the disease, numerous countries designated control zones and restrictions on live shellfish movements [5].

Both *B. ostreae* and *B. exitiosa* infect and multiply within haemocytes; in advanced infections, it becomes systemic, and the parasitic microcells can then be observed extracellularly between epithelial or interstitial cells in the gills and stomach or necrotic connective tissue areas [13,14]. Infection of *B. exitiosa* in *O. edulis* is also frequently observed in the gonads [15]. Clinical signs are not pathogen specific, do not always manifest, and can appear as shell gaping, eroded gills, yellow discolouration, and lesions—mostly hemocyte infiltration in the connective tissue of the gills, mantle, and digestive gland [14]. Seasonal outbreaks may result in high mortalities of oysters, especially after periods of stress such as spawning [16] or exposure to extreme temperatures [17]. Death usually occurs by exhaustion of energy reserves due to increasing haemocyte production and disruption of host tissues due to the high intensity of infection [18].

Infection with other microcell parasites, such as *Bonamia roughleyi*, *Mikrocytos mackini*, and possibly other unidentified haplosporidians, can cause a similar clinical appearance in oysters [3]. Current WOAH-approved diagnostic methods include histology and screening of heart smears or imprints for *B. ostreae* and *B. exitiosa* cells, followed by molecular confirmation of the parasite DNA [1,2]. Analysis of restriction fragment length polymorphism (RFLP) of PCR products and sequencing can be used for species identification [19]. Conventional PCR assays are available for both parasites, targeting the parasite *small subunit* (*SSU*) or *18S rRNA* gene [16,20,21]. Species-specific real-time qPCR assays are also available for both pathogens; these tests target either the *internal transcribed spacer* (*ITS*), the *SSU rRNA*, or one *actin* gene of the parasites [22,23,24,25,26]. Further protocols for detecting the protozoan in environmental samples based on PCR and Taqman^®^ qPCR have also been developed to study the parasite’s survival outside the host and transmission routes [27,28].

In the absence of effective treatments, the only effective measure is to prevent the introduction of the parasite to new areas by restricting live molluscan shellfish movement, prompt and effective outbreak control, and control of imports at border inspection posts (BIP). Loop-mediated isothermal amplification (LAMP) and recombinase polymerase amplification (RPA) are popular isothermal assays that allow for rapid and specific pathogen detection, with positive results typically under 30 min. Diagnostic tests based on these platforms have successfully been developed for the detection of shellfish pathogens such as *Perkinsus beihaiensis* [29], *Perkinsus* spp. [30], abalone herpesvirus (HaHV-1) [31], and ostreid herpesvirus (OsHV) [32,33,34]. Numerous isothermal assays are also available for other aquaculture pathogens [35]. These fast tests can be implemented with quick on-field sample preparation as magnetic solid-phase reversible immobilisation, lateral flow devices (LFD), microfluidics cartridges, or simple digestion buffers [36,37,38], offering great potential for their use as point-of-care tests (POCTs). The present study describes novel real-time LAMP tests for detecting *B. ostreae* and *B. exitiosa*. Two independent laboratories validated these new tests, and their accuracy was compared to a currently used diagnostic method [26]. These fast isothermal assays have great potential as POCTs for surveillance in BIPs and outbreak control.

## 2. Materials and Methods

### 2.1. DNA Controls

Control DNA for *B. exitiosa*, *B. ostreae*, and other species of parasites were sourced from the collection of the Ifremer ASIM Laboratory (France). The presence of the parasite on infected molluscs was confirmed by conventional PCR combined with sequencing [21,39,40], except for the samples infected with *M. mackini* and the one co-infected with *H. nesloni* and *H. costale*, which were only tested by real-time PCR. However, the status of those two samples was confirmed by the donor laboratory. Additionally, all samples were checked for the presence of *B. ostreae* and *B. exitiosa* using species-specific real-time PCR following protocols published by the EURL for mollusc diseases [26].

Samples used for the assessment of diagnostic accuracy consist of a set of 24 samples of flat oyster’s gDNA prepared from oysters originating from three different areas in France: Brest and Cancale bays in Brittany (*B. ostreae* endemic areas) and Sète Lagoon on the Mediterranean coast (where *B. exitiosa* has previously been detected). The status of each oyster was determined by conventional PCR [21] and Taqman PCR [26] from DNA extracted from the gills using the Wizard Genomic DNA Purification Kit from Promega^®^. Based on PCR results, DNA suspensions were pooled in order to obtain 2 pools of *B. ostreae*-positive gDNA representative of two levels of infection (named “Bo+” and “Bo++”), 2 pools of *B. exitiosa*-positive gDNA representative of two levels of infection (named “Be+” and “Be++”), 2 pools of *B. ostreae*- and *B. exitiosa*-positive gDNA (resulting from a mix of previous pools), and 4 pools of non-infected oyster DNA (Appendix A). Infection levels, marked either as positive “+” or strong positive “++”, differ from 3 Ct values approximatively (equivalent to a log in target concentration). Finally, each pool was checked by PCR before being aliquoted in 2 to 3 replicates in order to prepare sets of 24 samples.

### 2.2. LAMP Primers Assay Design

Three LAMP assays were *de novo* designed: two species-specific assays targeting one *actin* gene of either *B. ostreae* or *B. exitiosa*, and a third assay aiming to detect the *18S rRNA* gene of both *B. ostreae* and *B. exitiosa*. The LAMP primers were designed using the LAMP Designer 1.10 program (Premier Biosoft International v1.16). A partial sequence of the *B. ostreae actin-1* gene (BoAct1-14, GenBank accession no. AM410921.1) and the *B. exitiosa actin* gene (KM073107.1) were used for the primer design of each species-specific LAMP assay. A third sequence, the *B. ostreae 18S rRNA* gene (MZ305451.1), was used as a template for the primer design of the generic assay. Each LAMP assay consisted of two outer primers, F3 and B3, and two inner primers, FIP (which was made up of F1c and F2) and BIP (which was made up of B1c and B1). In addition, two loop primers (Loop-F and Loop-B) were incorporated to speed up the reaction. Multiple nucleotide alignments were conducted in MegAlign v7.0.21 (Lasergene DNASTAR) [41] to identify conserved and variable regions among parasites. The *in-silico* specificity of the primers was visualised against the nucleotide alignments of available sequences of actin and *18S rRNA* genes of *B. ostreae*, *B. exitiosa*, other *Bonamia* species, and the hosts, European and Chilean flat oysters (NCBI database accessed in September 2020). Pairwise comparisons among sequences were conducted to calculate the percentage of identity using the Maximum Composite Likelihood model [42]. The specificities of the primers were verified by a BLAST (Version 2.1.1.0) search [43].

### 2.3. LAMP Quantification DNA Standards

A fragment of 304 bp of the *B. ostreae actin-1* gene and a fragment of 155 bp of the *B. exitiosa actin* gene containing the LAMP product was amplified using the primers BO Actin Fw (5′-GGAGAAGATCTGGCACCACAG-3′) and BO Actin Rv (5′-CTCTTCCGGCGATGTCCAAT-3′) for *B. ostreae* and the set of primers LAMP F3 and B3 (Table 1) for *B. exitiosa*. PCR reactions were performed in a 50 µL reaction volume consisting of 1× green GoTaq^®^ Flexi buffer, 2.5 mM MgCl_2_, 1 mM dNTP mix, 1.25 units of GoTaq^®^ G2 Hot Start Polymerase (Promega), 50 pmol of each primer, and 2.5 µL of positive DNA for *B. ostreae* and *B. exitiosa*, respectively. After an initial denaturing step of 5 min at 95 °C, samples were subjected to 30 cycles of 30 s at 95 °C, 30 s at 55 °C, and 1 min at 72 °C, followed by a final extension step of 10 min at 72 °C in a Mastercycler nexus X2 (Eppendorf, Stevenage, UK). The PCR products were cloned into a pGem-T Easy plasmid vector (Promega, Hampshire, UK). The recombinant plasmid DNA was extracted using a QIAprep Spin Miniprep Column (Qiagen) following the manufacturer’s protocols. The template (dsDNA) copy number was calculated using a QuantiFluor dsDNA kit in a Quantus fluorimeter (Promega), and a plasmid dilution series from 10^6^ to 1 copy was conducted to generate standard curves for the B. ostreae and B. exitiosa actin LAMP assays. For the generic Bonamia LAMP assay, a 4 nmole Ultramer™ DNA oligo with the LAMP amplicon sequence (169 bp) was synthesised by Integrated DNA Technologies (IDT) and serially diluted to generate a standard curve.

### 2.4. LAMP Assay Protocol

Two independent laboratories (A and B) participated in evaluating the LAMP assays. Laboratory A conducted the tests in field portable Genie^®^ II or Genie^®^ III instruments (OptiGene) and laboratory B in a real-time thermocycler CFX96 (Biorad). Both laboratories used the same fast isothermal master mix (ISO-004, Optigene) and primer concentrations. LAMP reactions consisted of 15 μL of the fast isothermal master mix (ISO-004, OptiGene), 0.2 µM of each primer F3 and B3, 1.0 µM of each LoopF and LoopB, 2.0 µM of each FIP and BIP, and either plasmid standards or test DNA (2–5 µL), to a final volume of 25 μL. Tests were run over 30 min at the optimised temperature. Real-time monitoring of the isothermal amplification and the annealing temperature of the amplicon was measured as the change of fluorescence in the FAM channel. The reaction time (measured in minutes:seconds) was the time when fluorescence changed for the Genie instruments (time of positivity “Tp”) or the time to reach 1000 (t1000) relative fluorescence units (RFU) for the CFX96 thermocycler.

### 2.5. Optimization of the LAMP Assay Temperature

Laboratory A. To select the best isothermal reaction temperature for each of the three LAMP assays, a block gradient from 60 to 68 °C followed by an annealing step of 98–80 °C, ramping at 0.05 °C per second, was run on the Genie instruments with a positive DNA control consisting of 10^4^ copies of a recombinant plasmid containing the probing regions.

Laboratory B. The optimal temperature for the CFX96 instrument was verified by using gradient temperature from 63 °C to 68 °C for each LAMP assay. Tests were performed using control gDNA from oysters known to be infected with either *B. ostreae* or *B. exitiosa,* and both gDNA controls were used for the generic *Bonamia* LAMP assay.

In both laboratories, the reaction temperature that showed the faster amplification was then selected for further analyses.

### 2.6. LAMP Analytical Sensitivity

Analytical sensitivity of the LAMP assays was evaluated by testing serial dilutions of DNA standards in independent assays. The Limit of Detection (LoD) was determined as the lowest dilution producing consistent positive results.

The LoD of the LAMP assays was further compared with a species-specific Taqman Real Time PCR assay targeting the *actin-1* and *actin* genes of *B. ostreae* and *B. exitiosa* respectively following the protocol described on the EURL for Molluscs Diseases website [26]. The tests were performed using gDNA serial dilutions in triplicate in three independent runs.

### 2.7. LAMP Analytical Specificity

Specificity of the LAMP assays was evaluated by testing samples infected with *B. ostreae* or *B. exitiosa* originating from different geographic sites for assessing test inclusivity (to make sure the assay would not miss some targeted parasites) and with closely related parasites for assessing exclusivity (to make sure the assay would not amplify parasites other than parasites of the genus *Bonamia*). Additionally, a selection of LAMP products was examined with TapeStation Analysis Software A.02.02 using the Genomic DNA screen tape analysis (Agilent Technologies) to visualize the appearance of primer dimmers or other artefacts.

### 2.8. Diagnostic Accuracy of LAMP Assays for the Detection of Bonamiosis

LAMP assay accuracy was evaluated on a set of 24 flat oyster DNA samples of known status, including uninfected controls and samples infected with *B. ostreae*, *B. exitiosa*, or both parasites (see Section 2.1). The samples were analysed with LAMP assays independently in both laboratories A and B.

## 3. Results

### 3.1. Designing of LAMP Tests for the Detection of B. ostreae and B. exitiosa and Optimisation of the Isothermal Temperature Reaction

For each *Bonamia* species, species-specific fluorescence real-time LAMP assays were de novo designed, targeting either the *B. ostreae actin-1* gene or the *B. exitiosa actin* gene. A generic assay was designed on a conserved region of the *18S rRNA* gene to detect both species (Table 1, Figure 1). Pairwise comparisons of available sequences showed a nucleotide identity on the LAMP-targeted region of 100% for the *B. ostreae actin-1* gene and 96.8% for the *B. exitiosa actin* gene. The nucleotide identities of the homologous *actin* gene between both bonamia species were 91.2% and 84% for the *B. ostreae* and *B. exitiosa* LAMP assays, while the hosts *O. edulis* and *O. chilensis* showed nucleotide identities with the parasites ’*actin* gene of ~80%. The nucleotide identity of the *18S rRNA* gene among bonamia species was ~97%. The same region in other haplosporidian species showed a nucleotide identity of ~87% (partial gene alignments can be found in Appendix A).

For each LAMP assay, the optimal isothermal reaction temperature was tested independently by the two participant laboratories. Laboratory A determined the fastest reaction temperature at 67.6, 67.3, and 65.4 °C for each of the *B. ostreae*, *B. exitiosa*, and generic *Bonamia* LAMP assays when using a Genie instrument. Laboratory B observed fast amplification with temperature ranging from 63 to 65 °C (t1000 was between 7.5 and 7.7 min) for the three LAMP assays, while temperatures higher than 65 °C showed slower reactions (t1000 = 9 min) when using a CFX96 thermocycler. For further tests, 65 °C was selected as the optimal reaction temperature in laboratory B (Figure 2). In both laboratories, the melting temperature of specific LAMP products were in the range of 89.8–90.2 °C, 87.0–87.6 °C, and 86.2–86.6 °C for each of the *B. ostreae*, *B. exitiosa*, and the generic *Bonamia* LAMP assays (examples of inclusivity tests are provided in Appendix A).

### 3.2. LAMP Analytical Sensitivity

In both laboratories, the analytical limit of detection (LoD) with a reliable detection of all replicates amplified under 30 min was 50 copies/µL for each of the three LAMP assays (Table 2). Detection of up to 10 copies/µL for the *B. ostreae* and *Bonamia* generic assays and up to 1 copy/µL for the *B. exitiosa* assay was occasionally observed; however, it was unreliable between replicates and, therefore, considered below the LoD. Longer runs (up to 1 h) did not increase the LoD. The *B. exitiosa* LAMP assay showed the fastest detection, with amplifications recorded from 5:15 ± 0:12 to 10:30 ± 0.30 (minutes:seconds) for detecting 10^4^ to 50 copies/µL. The time of positivity (Tp) of the *Bonamia* sp. LAMP assay ranged from 9:30 ± 0.01 to 21:52 ± 0:16 min:s, while the *B. ostreae* assay showed the slowest Tp, ranging from 16:30 ± 2:10 to 27:20 ± 3:30 min:s (Figure 3). A linear correlation between the Tp and the gene copy number showed a regression coefficient between −2.7 and −3.8 and an R^2^ of 0.8–0.9 (Figure 3D). A comparison of the LAMP sensitivity with the sensitivity observed in a real-time PCR showed that the real-time PCR and the generic *Bonamia* LAMP assay detected more replicates at high dilutions of positive controls (Table 3). The results suggest that the species-specific LAMP assays are approximately 1 log less sensitive than the generic *Bonamia* LAMP assay, which has a similar sensitivity to the real-time PCR.

### 3.3. LAMP Analytical Specificity

The *B. ostreae* LAMP assay only amplified *B. ostreae* samples without cross-reactions with *B. exitiosa* or other closely related parasites. Similarly, the *B. exitiosa* LAMP assay only amplified *B. exitiosa* samples but gave inconsistent results detecting a particular *B. exitiosa* sample (false negative). The generic *Bonamia* LAMP assay amplified all tested *B. ostreae* and *B. exitiosa* samples with no amplification of other closely related parasites except for a highly infected *Haplosporidium costale* sample that produced doubtful results (late amplification). Two other *H. costale* samples with lower parasitic load did not generate false positives (Table 4). Analysis of the LAMP products visualised on a TapeStation did not indicate primer dimers or any other artefact (Appendix A).

### 3.4. LAMP Diagnostic Accuracy for the Detection of Bonamiosis

The LAMP assays amplified positive samples within 7 and 20 min without cross-reaction with uninfected samples. Both the *B. ostreae* and generic *Bonamia* LAMP results showed 100% agreement with the sample status (100% diagnostic accuracy). However, the *B. exitiosa* LAMP test failed to detect one positive sample in each laboratory; thus, the diagnostic accuracy for the *B. exitiosa* LAMP test was considered ~95.8% (Table 5, Appendix A).

## 4. Discussion

This is the first description of fluorescence real-time LAMP assays for the fast detection of bonamiosis in oysters. The assays described herein were evaluated independently in two laboratories using different instruments. Infection with haplosporidian parasites has led to historical declines in oyster populations; in particular, the European flat oyster beds were devastated by the introduction of *B. ostreae* [44]. Recently, the introduction of *B. exitiosa* in areas of high prevalence of *B. ostreae* has highlighted the need for species-specific diagnostic methods [45].

The current study selected *actin* genes of *B. ostreae* and *B. exitiosa* to design species-specific LAMP assays. Similarly, *Bonamia actin* genes have been evaluated successfully for developing species-specific Taqman qPCRs [24,26]. Here, the in silico analysis of the LAMP primers binding sites and the LAMP tests conducted with reference material showed 100% specificity of each species-specific LAMP assay with their target parasite; however, the lack of published sequences for *actin* genes of other *Bonamia* species, such as *Bonamia perspora,* infecting the flat oyster *Ostreola equestris* prevented further validation of the in silico specificity of the primers [46].

To allow for the detection of both *Bonamia* species, a generic assay was also designed in a conserved region of the *18S rRNA* gene. The in silico analysis of primer binding sites showed conserved sequences among *Bonamia* species on the *18S rRNA* gene with minor single-nucleotide polymorphisms (SNPs) included in the primer design. A sequence of *Bonamia* sp. infecting *Ostrea sandvicencsis* (JF831803.1) showed a higher number of SNPs that were not possible to incorporate into the degenerated primers. Thus, the ability of the generic *Bonamia* assay to detect this species could be affected. Phylogenetic analysis based on the *SSU rRNA* gene placed this *Bonamia* species as highly divergent, with just 90–91% nucleotide similarity with published sequences of *B. exitiosa*, *B. ostreae*, and *B. perspora*, and suggested it as a new species [4].

The specificity of the LAMP assays was further tested against a range of close relative pathogens. Both species-specific LAMP assays were 100% specific to their target species; however, the generic *Bonamia* assay showed an inconsistent cross-reaction with *H. costale* DNA, observed as a late amplification in a sample showing a high infection level. The nucleotide differences observed in the *H. costale 18S* rRNA and the LAMP primers (Appendix A) did not prevent unspecific binding of the primers, the presence of unknown parasite(s) causing cross-reaction cannot be discarded for this particular sample. Despite the worldwide economic impact of haplosporidian parasites in aquatic organisms, the genome of these microcells has yet to be characterised. Sequences publicly available are just segments of the parasites *SSU* or *18S rRNA*, internal transcribed spacer regions (*ITS1-5.8S-ITS2*), and *actin* genes [47] (GenBank last accessed on 23/02/2023). Thus, the lack of published genomes for *Bonamia* species and other haplosporidian parasites jeopardises the design of primers and the selection of target genes. Further sequencing data and reference material on *Bonamia* species will aid in validating and refining these rapid and promising diagnostic assays.

The three LAMP assays showed a specific and distinct melting temperature of their LAMP product during the specificity tests. Although the incidence of false positive results in LAMP assays has been reported before [48], the real-time fluorescence detection of the LAMP amplification and the analysis of the product melting temperature allows for better discrimination of non-specific amplifications [49].

In both laboratories, the analytical sensitivity of the LAMP assays was 50 copies/µL in a 30 min run. Other LAMP assays have achieved lower LoD with longer runs, such as a real-time turbidimeter LAMP assay for the detection of *Perkinsus* spp that allowed the detection of up to 10 copies of a plasmid DNA at 49.8 min [29] and a colourimetric LAMP assay that detected up to 20 copies of OsHV-1 in a 60 min run [32]. In the current study, longer runs did not increase the LoD, and although 10 and 1 copies/µL were occasionally detected, it was inconsistent between replicates and therefore considered below the LoD.

Despite the excellent performance displayed by the species-specific LAMP assays, their LoD was still ~1 fold less sensitive than a species-specific real-time PCR targeting the same actin gene with a reported LoD between 1 and 2.5 copies/µL [24,50]. Interestingly, when using gDNA dilutions rather than a standard template, the analytical sensitivity of the generic *Bonamia* LAMP assay, which targets the parasite *18S rRNA* gene, was ~1 fold more sensitive than the *actin* LAMP assays and similar to the real-time PCR. The fact that the copy number of the *18S rRNA* gene per cell is likely higher than the copy number of the *actin* gene [51] might contribute to the higher sensitivity of the generic *Bonamia* LAMP test. Thus, although the species-specific LAMP assays developed in this study might not positively identify samples with very low levels of parasites otherwise identified by real-time PCR and histopathology [22,23], the parallel use of the generic *Bonamia* LAMP assay together with the species-specific assays might aid in identifying samples infected with very few parasites.

Lastly, when evaluated on a set of 24 samples of known status, both *B. ostreae* and generic *Bonamia* LAMP assays showed 100% diagnostic accuracy. However, the *B. exitiosa* LAMP assay accuracy was ~95.8%, with errors consisting of false negative results. Testing the samples with both species-specific and the generic LAMP assay might aid in identifying those false negatives. Although a multiplex LAMP assay for the simultaneous detection of both *B. ostreae* and *B. exitiosa* was not addressed in this study, the fact that both assays have a similar optimal reaction temperature (67 °C in a Genie instrument and 65 °C in a thermocycler) allows for running both assays on the same strip or plate with insignificant differences in their time of positivity. Per instant, simultaneous real-time fluorescence LAMP detection of bacteria and viruses of concern in aquaculture systems has been achieved using microfluidic chips integrated with LAMP (on-chip LAMP) [52].

In summary, two independent laboratories have evaluated real-time LAMP assays for detecting *Bonamia* species. The results presented by both laboratories confirm the good performance of these assays for detecting bonamiosis in oysters. The main advantage is the rapidity and affordability of the assays, with positive results in under 30 min. Moreover, the LAMP chemistry is tolerant to known PCR inhibitors, permitting a less stringent DNA extraction [53] and, thus, potentially allowing its deployment as POC tests. Although we have not validated these assays for the quantitative detection of the parasite, they are a valuable tool for rapid confirmation of the infection with a positive or negative result to inform on the presence/absence of the parasite. The main downside of these assays is the LoD, which detects up to 50 copies/µL of the target gene and, thus, might not be as sensitive as a real-time PCR. These assays are not intended to replace a centralised confirmatory diagnostic. Still, they can be a valuable tool for a presumptive diagnostic for field suspicion and surveillance programs at border inspection posts and could aid in monitoring oyster population health and early detection of parasite introduction in natural habitat restoration, such as the European Oyster Reef Restoration program “NORA”.

## Figures and Tables

**Figure 1 pathogens-13-00132-f001:**
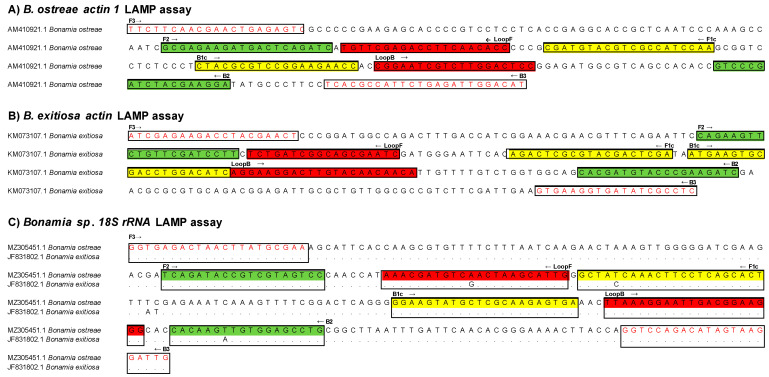
Partial nucleotide sequences of (**A**) *Bonamia ostreae actin-1* gene (GenBank accession number AM410921.1), (**B**) *Bonamia exitiosa actin* gene (KM073107.1), and (**C**) *B.* ostreae *and B. exitiosa 18S rRNA* genes (MZ305451.1 and JF831802). The LAMP primer binding sites are highlighted as follows: outer primers F3 and B3 boxed in white; inner primers FIP (F1c + F2) and BIP (B1c + B2) boxed in yellow and green, respectively; loop primers LoopF and LoopB boxed in red.

**Figure 2 pathogens-13-00132-f002:**
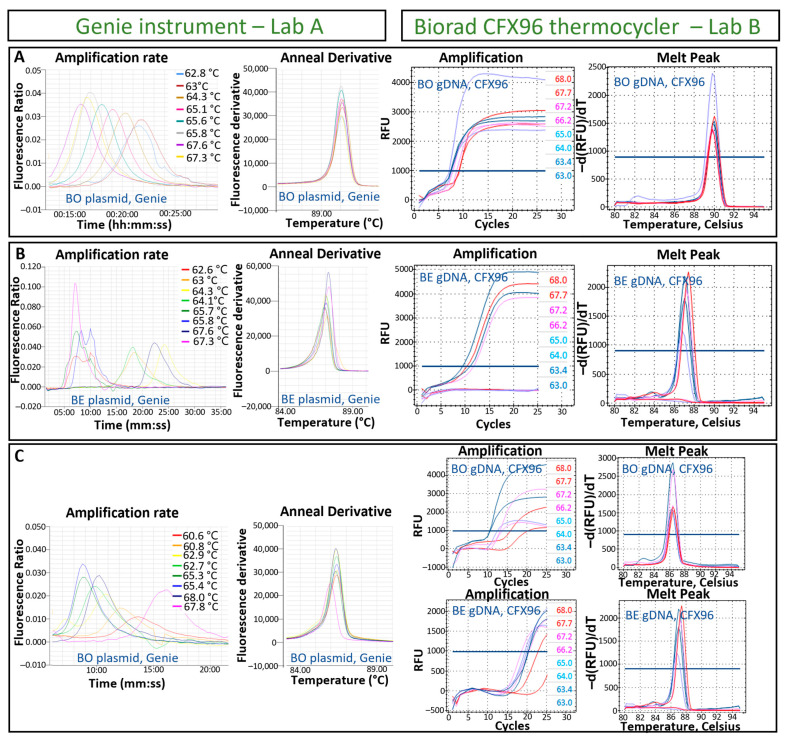
Optimisation of the loop-mediated isothermal amplification (LAMP) reaction temperature for the species-specific detection of (**A**) *Bonamia ostreae* (BO), (**B**) *B. exitiosa* (BE), and (**C**) generic assay for the detection of both *B. ostreae* and *B. exitiosa***.** The amplification and anneal derivative (melt peak) for each LAMP assay are shown when running the assay in either a Genie instrument using plasmid DNA or a CFX96 thermocycler using gDNA.

**Figure 3 pathogens-13-00132-f003:**
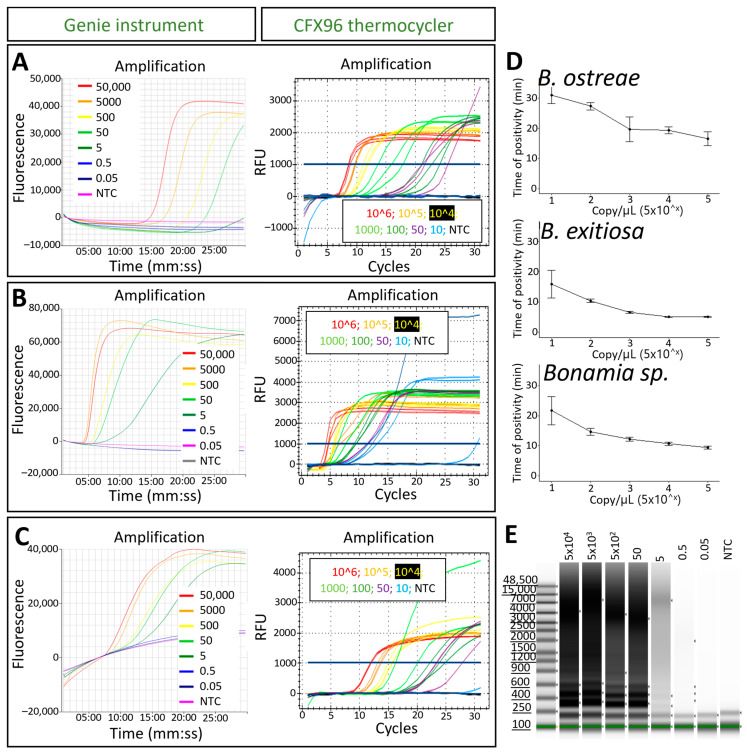
Analytical sensitivity of the species-specific (**A**) *Bonamia ostreae* and (**B**) *B. exitiosa* and (**C**) the generic *Bonamia* LAMP assays. The isothermal amplification of serial dilutions of a recombinant plasmid control is shown either in a Genie instrument (**left**) or a CFX96 thermocycler (**right**). (**D**) Plots showing the linear correlation of copy number/µL and the time of positivity (minute average ± standard deviation). (**E**) Gel image run in a TapeStation of LAMP products from serial dilutions of a standard control obtained with the generic *Bonamia* LAMP assay.

**Table 1 pathogens-13-00132-t001:** Primer sequences for the loop-mediated isothermal amplification (LAMP) assays of *Bonamia ostreae actin-1* (GenBank accession number AM410921.1), *B. exitiosa actin* (KM073107.1), and *B. ostreae 18S rRNA* (MZ305451.1) genes.

LAMP Assay	Primer	Primer Sequence (5′-3′)	Position	Tm °C	GC%	Amplicon *
*B. ostreae actin-1*	BO_F3	TTCTTCAACGAACTGAGAGTC	37	59.9	42.9	158 bp
	BO_B3	ATGTCCAATCTCAGAATGGC	308	60	45	
	BO_FIP (F1c + F2)	TTGGATGGCGACGTACATCGGCGAGAAGATGACTCAGATC				
	BO_BIP (B1c + B2)	CTACGCGTCCGGAAGAACCTCCTTCGTAGATCGGGAC				
	BO_LoopF	GGTGTTGAAGGTCTCGAACA	156	62.3	50	
	BO_LoopB	CGGAATCGTCTTGGACTCC	216	62.2	57.9	
*B. exitiosa actin*	BE_F3	ATCGAGAAGACCTACGAACT	472	60	45	155 bp
	BE_B3	GAGGCGATATCACCTTCAC	763	59.8	52.6	
	BE_FIB (F1C + F2)	TCGAGTCGTACGCGAGTCTCAGAAGTTCTGTTCGATCCTT				
	BE_BIP (B1c + B2)	ATGAAGTGCGACCTGGACATCGATCTTCGGGTACATCGTG				
	BE_LoopF	GATTCGCTGCCGATCAGA	578	62	55.6	
	BE_LoopR	AGGAAGGACTTGTACAACAACA	634	61.9	40.9	
*B. ostreae 18S*	B_F3	GGTGAGACTAACTTATGCGAA	881	59.8	42.9	169 bp
	B_B3	CAATCCTTACTATGTCTGGACC	1185	60	45.5	
	B_FIB (F1C + F2)	AGTGCTGAGGAAGTTTGRTAGCTCAGATACCGTCGTAGTCC				
	B_BIP (B1c + B2)	GGAAGTATGCTCGCAAGAGTGACAGGCTCCACAWCTTGTG				
	B_LoopF	CAATGCTTAGTYGACATCGTTT	1007	61.6	40.9	
	B_LoopR	CTTAAAGGAATTGACGGAAGGG	1086	61.3	45.5	

* Initial amplicon size, which is amplified and concatenated. Tm: primer melting temperature. GC%: percentage content of guanine–cytosine.

**Table 2 pathogens-13-00132-t002:** Analytical sensitivity of the species-*specific Bonamia ostreae* and *B. exitiosa* LAMP assays and the generic assay for detecting both species.

LAMP Assay	*Species-Specific* *B. ostreae*		*Species-Specific* *B. exitiosa*		*Generic Bonamia*	
Copies/µL	A	B	A	B	A	B
10^6^	4/4	6/6	4/4	6/6	4/4	6/6
10^5^	4/4	6/6	4/4	6/6	4/4	6/6
10^4^	4/4	6/6	4/4	6/6	4/4	6/6
10^3^	4/4	6/6	4/4	6/6	4/4	6/6
10^2^	4/4	6/6	4/4	6/6	4/4	6/6
50	4/4	6/6	4/4	6/6	4/4	6/6
10	2/4	2/6	4/4	4/6	3/4	2/6
1	0/4	0/6	0/4	1/6	0/4	0/6
NTC	0/4	0/6	0/4	0/6	0/4	0/6

Sensitivity was estimated by testing serial dilutions of recombinant plasmid controls. Tests were conducted independently in laboratories A and B. The number of positives versus total tests is shown for each concentration. NTC: non-template control.

**Table 3 pathogens-13-00132-t003:** Sensitivity comparison of LAMP assays versus Taqman real-time PCR using serial dilutions of *B. ostreae* and *B. exitiosa* gDNA in a CFX96 thermocycler.

** *B. ostreae* **	**LAMP *B. ostreae***		**LAMP Generic *Bonamia***		**Taqman qPCR**
**gDNA Dilution**	**Positive/Total**	**TM °C**	**Positive/Total**	**TM °C**	**Positive/Total**
1/10	9/9	90	9/9	86.4–86.6	9/9
1/100	9/9	90	9/9	86.4	9/9
1/1000	9/9 (1 doubtful)	90	9/9	86.4–87	9/9
1/10,000	0/9	none	8/9 (1 doubtful)	86.2–86.4	6/9
1/100,000	0/9	none	1/9	86.2	0/9
1/1,000,000	0/9	none	0/9	none	0/9
NTC	0/9	none	0/9	none	0/9
** *B. exitiosa* **	**LAMP *B. exitiosa***		**LAMP Generic *Bonamia***		**Taqman qPCR**
**gDNA Dilution**	**Positive/Total**	**TM °C**	**Positive/Total**	**TM °C**	**Positive/Total**
1/10	9/9	87.2–87.6	9/9	86.0–86.6	9/9
1/100	9/9 (1 doubtful)	86.2–87.0	9/9	86.2–86.4	9/9
1/1000	5/9 (1 doubtful)	86.4–88.0	9/9	86.0–86.2	9/9
1/10,000	1/9	87.6	2/9 (1 doubtful)	86.0–86.2	8/9
1/100,000	0/9	none	2/9 (1 doubtful)	86.0–86.2	1/9
1/1,000,000	0/9	none	0/9	none	0/9
NTC	0/9	none	0/9	none	0/9

The number of positives versus total tests is shown for each dilution. NTC: non-template control. TM: amplicon melt temperature. Doubtful: late amplification that did not reach the 1000 RFU threshold.

**Table 4 pathogens-13-00132-t004:** Analytical specificity of the species-specific *Bonamia ostreae* and *B. exitiosa* and the generic *Bonamia* LAMP assays.

				LAMP *B. ostreae*		LAMP *B. exitiosa*		LAMP *Bonamia* sp.	
	Pathogen	Host	Origin	Results	TM	Results	TM	Results	Tm
Exclusivity	*Marteilia type O*	*Ostrea edulis*	France (Ifremer)	−	None	−	None	−	None
	*Mikrocytos veneroïdes*	*Donax trunculus*	France (Ifremer)	−	87.00 *	−	None	−	None
	*Haplosporidium costale*	*Magallana (ex Crassostrea) gigas*	France (Ifremer)	−	None	−	None	+/−	86.80 **
	*Haplosporidium nelsoni &* *H. costale*	*Crassostrea virginica*	USA (VIMS)	−	None	−	None	−	None
	*Mikrocytos mackini*	*Magallana (ex Crassostrea) gigas*	Canada (DFO)	−	None	−	None	−	None
Inclusivity	*Bonamia exitiosa*	*Ostrea edulis*	France (Ifremer)	−	None	+	87.40	+	86.2
	*Bonamia exitiosa*	*Ostrea edulis*	France (Ifremer)	−	None	+	87.60	+	86.2
	*Bonamia exitiosa*	*Ostrea edulis*	Turkey (VCRI)	−	None	−/+	87.40	+	86.2
	*Bonamia ostreae*	*Ostrea edulis*	France (Ifremer)	+	89.80	−	None	+	86.4
	*Bonamia. ostreae*	*Ostrea edulis*	France (Ifremer)	+	90.00	−	None	+	86.4
	*Bonamia ostreae*	*Ostrea edulis*	France (Ifremer)	+	89.80	−	None	+	86.4

The asterisk denotes a doubtful result as follows: (*) late amplification (t1000 = 30 min) with nonspecific amplicon melt temperature (TM); (**) late amplification (t1000 = 26 min) but TM within the expected dynamic range.

**Table 5 pathogens-13-00132-t005:** LAMP detection of *Bonamia ostreae* and *B. exitiosa* using species-specific and a generic LAMP assay on a set of 24 gDNA samples.

Sample	Status	LAMP *B. ostreae*		LAMP *B. exitiosa*		LAMP *Bonamia* sp.	
		Lab A (Tp)	Lab B (TM)	Lab A (Tp)	Lab B (TM)	Lab A (Tp)	Lab B (TM)
1	Bo+ Be++	+(12:00)	+(90.00)	+(11:00)	+(87.60)	+(11:59)	+(86.40)
2	negative	–	–	–	–	–	–
3	Be+	–	–	+(17:30)	+(87.40)	+(12:14)	+(86.20)
4 *	Be++	–	–	–	+(87.40)	+(11:44)	+(86.20)
5	negative	–	–	–	–	–	–
6	Bo++	+(12:00)	+(90.00)	–	–	+(9:59)	+(86.20)
7	negative	–	–	–	–	–	–
8	Be+	–	–	+(12:45)	+(87.60)	+(13:15)	+(86.20)
9 *	Be+	–	–	+(17:30)	–	+(12:30)	+(86.20)
10	Bo+	+(8:30)	+(90.00)	–	–	+(9:15)	+(86.40)
11	Be++	–	–	+(15:15)	+(87.60)	+(10:00)	+(86.20)
12	negative	–	–	–	–	–	–
13	negative	–	–	–	–	–	–
14	Bo++ Be+	+(16:45)	+(90.00)	+(12:00)	+(87.40)	+(10:15)	+(86.40)
15	Bo++ Be+	+(9:00)	+(90.00)	+(13:40)	+(87.40)	+(10:00)	+(86.00)
16	Bo+	+(8:30)	+(90.20)	–	–	+(10:30)	+(86.40)
17	Be++	–	–	+(12:00)	+(87.60)	+(9:30)	+(86.20)
18	Bo+ Be++	+(9:00)	+(90.00)	+(10:15)	+(87.60)	+(13:00)	+(86.40)
19	Bo++	+(7:45)	+(90.00)	–	–	+(9:30)	+(86.40)
20	negative	–	–	–	–	–	–
21	negative	–	–	–	–	–	–
22	negative	–	–	–	–	–	–
23	Bo+	+(8:15)	+(90.00)	–	–	+(10:15)	+(86.40)
24	Bo++	+(9:00)	+(90.00)	–	–	+(9:15)	+(86.40)

Tests were conducted independently in laboratories A and B. Tp: time of positivity (minutes:seconds) of the LAMP assay measured with a Genie Instrument. TM: amplicon melt temperature of the LAMP product measured with a thermocycler. Sample status is defined as either a negative or a *B. ostreae* (Bo) or *B. exitiosa* (Be) positive and categorised as positive (+) or strong positive (++) depending on the parasite load. Asterisks (*) denote samples with a disagreement between the test result and the sample status.

## Data Availability

The data presented in this study is contained within this article or its Appendix A.

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
