# Peer review of "Loop-Mediated Isothermal Amplification for the Fast Detection of Bonamia ostreae and Bonamia exitiosa in Flat Oysters"

_pathogens, 2024, doi:10.3390/pathogens13020132_

Round 1

Reviewer 1 Report

Comments and Suggestions for Authors

The haplosporidian parasites Bonamia ostreae and B. exitiosa are serious oyster pathogens that are quite widespread and apparently continue to spread to new regions, posing a significant threat to both native oyster settlements and oyster farms.

Despite the fact that the diagnosis and species identification of these parasites are rather well developed both at the level of microscopy, histology and on the basis of DNA sequences, the development of new methods of rapid diagnostics and improvement of their accuracy is an urgent task. The present manuscript is devoted to the solution of this problem, therefore its publication is relevant and of interest both for researchers and practitioners.

In general, the study is performed at a good methodological level, it is described in detail, the results are representative, and the article corresponds to the journal.

I have no significant comments on the work.

A few minor remarks are as follows.

1. The structure of the manuscript is somewhat confusing, as the Material and Methods section is placed after the Discussion. I think it should be moved before Results.

2. In addition, in the Material and Methods section it would be useful to present information from which nature area, possibly farm, the infected and uninfected oysters were taken, as it is written that the infection was natural.

3. How were samples taken: pieces of tissue? from what organs? or is it a homogenate from all the tissues of the mollusc? Add this information.

4. Was the degree of infection of the oysters used in the experiments assessed? In the title to Table 5 it is written that they "are categorised as: + and ++ depending on the parasite load". How was the "parasite load" determined? Was it based on the results of the test being checked or preliminarily, if preliminarily, how? and what level of the "parasite load" corresponds to+ and ++. Add this information.

Author Response

Dear Reviewer,

Thank you for your time and consideration in reviewing the manuscript.

(1) We have moved the M&M section where it corresponds.

(2) Thank you for highlighting this. We have added the missing information in a new paragraph in section 2.1, from lines 100 to 113. The animals were taken from three naturally infected areas in France: Brest and Cancale bays in Brittany (B. ostreae endemic areas) and Sète lagoon on the Mediterranean coast.

(3) gDNA was extracted from gill tissues; we have added this information in the new line 103.

(4) Infection was determined by conventional PCR and Taqman qPCR to categorise the samples by their level of infection in negative, positive (+), or strong positive (++). We have added new information on how we pooled the samples in lines 104-111 and a new table in Supplement Table S1.

Reviewer 2 Report

Comments and Suggestions for Authors

The manuscript 'Loop-mediated Isothermal Amplification for the Fast Detection of Bonamia ostreae and Bonamia exitiosa in Flat Oysters' provides a solid contribution to marine pathogen diagnostics. The study's focus on developing LAMP assays for bonamiosis in oysters addresses a critical need in aquaculture. The methodology, targeting both species-specific actin genes and a generic 18S rRNA gene, is well thought out and demonstrates the authors' understanding of molecular diagnostics.

The reported results, showing high accuracy for both B. ostreae and B. exitiosa, are promising, especially considering the rapid detection time and sensitivity of the assays. These aspects are crucial for effective disease management in oyster populations.

Overall, this study is a significant leap forward in aquatic pathogen diagnostics, offering a practical, rapid, and highly accurate method for detecting Bonamia species. The research is methodologically sound, well-executed, and presented in a clear and insightful manner. It stands as a valuable contribution to marine biology and holds great promise for improving disease management practices in aquaculture.

Author Response

Dear Reviewer,

Thank you for your time reviewing this manuscript and consideration,

Kindest regards

Authors